# Influence of the Gas Preheating Temperature on the Microstructure and Electrical Resistivity of Copper Thin Films Prepared via Vacuum Cold Spraying

Kai Ma, Qing-Feng Zhang, Hui-Yu Zhang, Chang-Jiu Li and Cheng-Xin Li *

State Key Laboratory for Mechanical Behavior of Materials, School of Materials Science and Engineering, Xi'an Jiaotong University, Xi'an 710049, China
* Correspondence: licx@mail.xjtu.edu.cn

**Abstract:** Vacuum cold spraying (VCS) has emerged as an environmentally sustainable method for fabricating ceramic and metal films. A high particle impact velocity is a critical factor in the deposition of metal particles during the VCS process, which can be significantly enhanced through gas preheating. This study employs Computational Fluid Dynamics (CFD) simulations to investigate the substantial impact of gas preheating temperature on particle impact velocity and temperature. Elevating the gas temperature leads to higher particle impact velocity, resulting in severe deformation and the formation of dense copper films. The experimental results indicate improvements in both film compactness and electrical properties with gas preheating. Remarkably, the electrical resistivity of the copper film deposited at a gas preheating temperature of 350 °C was measured at $4.4 \times 10^{-8}$ Ω·m. This study also examines the evolution of cone-shaped pits on the surface of copper films prepared on rough substrates. VCS demonstrates a self-adaptive repair mechanism when depositing metal films onto rough ceramic substrates, making it a promising method for ceramic surface metallization.

**Keywords:** vacuum cold spraying (VCS); aerosol deposition (AD) method; copper thin films; gas preheating; electrical resistivity

## 1. Introduction

In recent years, there has been a rapid development in microelectromechanical systems (MEMS) products, which has garnered significant attention in relation to the design and fabrication of miniaturized micro–nano devices. Metal materials, specifically copper and silver, have gained widespread utilization in the manufacturing of electrical and heat-dissipating components due to their remarkable electrical and thermal conductivity which is due to their exceptional electrical and thermal conductivity [1,2]. Consequently, there is an urgent need to investigate metallization processes to meet the evolving demands of micro–nano device manufacturing technology. Up to this point, methods such as electroplating [3], chemical deposition [4], chemical vapor deposition [5], and magnetron sputtering [6] have commonly been employed for metallization processes. Nevertheless, these methods present certain environmental concerns and exhibit limitations in terms of thick film deposition efficiency. To address these challenges, it is imperative to develop a more straightforward, efficient, and environmentally friendly metallization process for manufacturing purposes.

Vacuum cold spraying (VCS), also known as the aerosol deposition (AD) method [7,8], has attracted attention as a dry film manufacturing process technology. It enables the rapid preparation of ceramic films, ranging in thickness from several to tens of microns, on various substrates such as ceramics, metals, glass, and even polymers [9–14]. Currently, there are limited reports on the deposition of metal coatings using AD, but these reports often exhibit higher electrical resistivity [15–17]. The formation of the ceramic films is attributed to the reduction of the crystallite size caused by a fracture and plastic deformation in the VCS process [7,18]. Given the ductile nature of metal materials in comparison to brittle ceramics,

it is anticipated that the deposition mechanism of metal particles significantly differs from that of ceramic particles. In cold spraying, the bonding of metal particles is due to plastic deformation and adiabatic shear instability occurring at high impact velocities [19]. It has been confirmed that the impact velocity and temperature of the particles are the most critical factors for the deposition of metal particles [20,21]. However, the high impact velocity is attributed to the high gas pressure of tens of atmospheres and the high gas temperature of hundreds of Celsius. Due to the low gas pressure of one atmosphere or even lower and room temperature gas used in the VCS process, these pose a formidable challenge to achieving metal films as dense as those produced via cold spraying [17,22]. The increase in gas temperature will not only effectively increase the particle impact velocity, but also enable in situ particle heating [23,24]. Thus, elevating the gas temperature emerges as an effective strategy to enhance the preparation of copper thin films via VCS.

In this study, the influence of gas temperature on gas flow and particle acceleration was investigated through Computational Fluid Dynamics (CFD) simulations. The copper thin films were fabricated on alumina substrates using the VCS process, comparing cases with and without gas preheating. The electrical resistivity and microstructure of the copper films were characterized. This study not only contributes to a deeper understanding of the VCS technique but also provides a foundation for the design and development of high-performance copper thin films in advanced electronic and microelectronic devices.

## 2. Materials and Methods

Commercial copper powder (Shanghai St-nano Science and Technology Co., Ltd., Shanghai, China) was used as feedstock material for the experiments. The copper powder particles, with mean particle sizes of 1.8 μm, exhibit a spherical morphology and are devoid of porosity, as shown in Figure 1. Alumina slides (12 × 12 × 2 mm) with a surface roughness of Ra 1.0 μm were used as substrates. Before spraying, the substrates was ultrasonically cleaned in ethanol to obtain a clean surface.

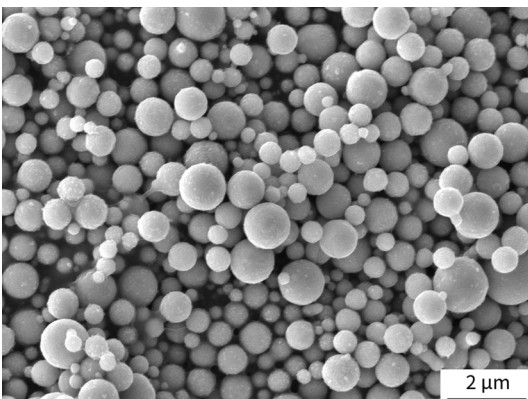

**Figure 1.** The morphology of the copper powder.

The spray process was conducted utilizing a home-developed vacuum cold spraying system by Xi'an Jiaotong University, as previously detailed [25–27]. Notably, a redesigned de Laval nozzle, featuring a throat diameter of 0.7 mm and an exit diameter of 2.5 mm, was utilized for this study. Helium (99.99%) was employed as the carrier gas, flowing at a rate of 5 L/min. To modulate the gas temperature, a home-made heating device was positioned between the gas pipe and the nozzle in the vacuum deposition chamber. This device allowed for control of the gas temperature, ranging from 20 °C to 350 °C. Diverse thicknesses of copper films were achieved through variations in the number of scanning passes. A comprehensive list of the key process parameters employed in the VCS process is provided in Table 1.

**Table 1.** Deposition parameters of the vacuum cold spraying.

| Parameter | Unit | Value |
|---|---|---|
| Gas flow rate | L/min | 5 |
| Chamber pressure | Pa | <300 |
| Distance from nozzle exit to the substrate | mm | 5 |
| Nozzle traversal speed | mm/s | 2 |
| Gas temperature | °C | 20–350 |

After the vacuum cold spraying experiments, the top view and cross-sectional microstructures of the copper films deposited at different temperatures were examined using field-emission scanning electron microscopy (FE-SEM, MIRA3 LMH, TESCAN, Brno-Kohoutovice, Czech Republic). The resistivity of the VCS-deposited copper films was measured using a 4-point probe (RTS-9, Four Probes Technology Co., Ltd., Shenzhen, China).

A commercially available CFD code, Fluent (17.0 Fluent Inc., New York, NY, USA), was used to predict steady gas flow and analyze particle acceleration behavior during VCS. Thanks to the axisymmetric nozzle used in this study, a two-dimensional axisymmetric model, as shown in Figure 2a, was established to save computational time [28,29]. The dimension of the nozzle in this simulation was consistent with the experimental nozzle used. The substrate was placed 5 mm away from the nozzle exit. The computational domain was meshed by structured grids with 23,000 nodes to achieve a grid-independent solution, as shown in Figure 2b,c. The gas inlet was chosen as the pressure inlet with a pressure value of 0.05 MPa and four temperature values of 293 K, 423 K, 523 K, and 623 K, while the outlet pressure and outlet temperature were constant and equal to 300 Pa and 300 K. The reference atmosphere was at a pressure of 0 Pa, and the heat transfer process between the nozzle wall and the gas flow was not considered.

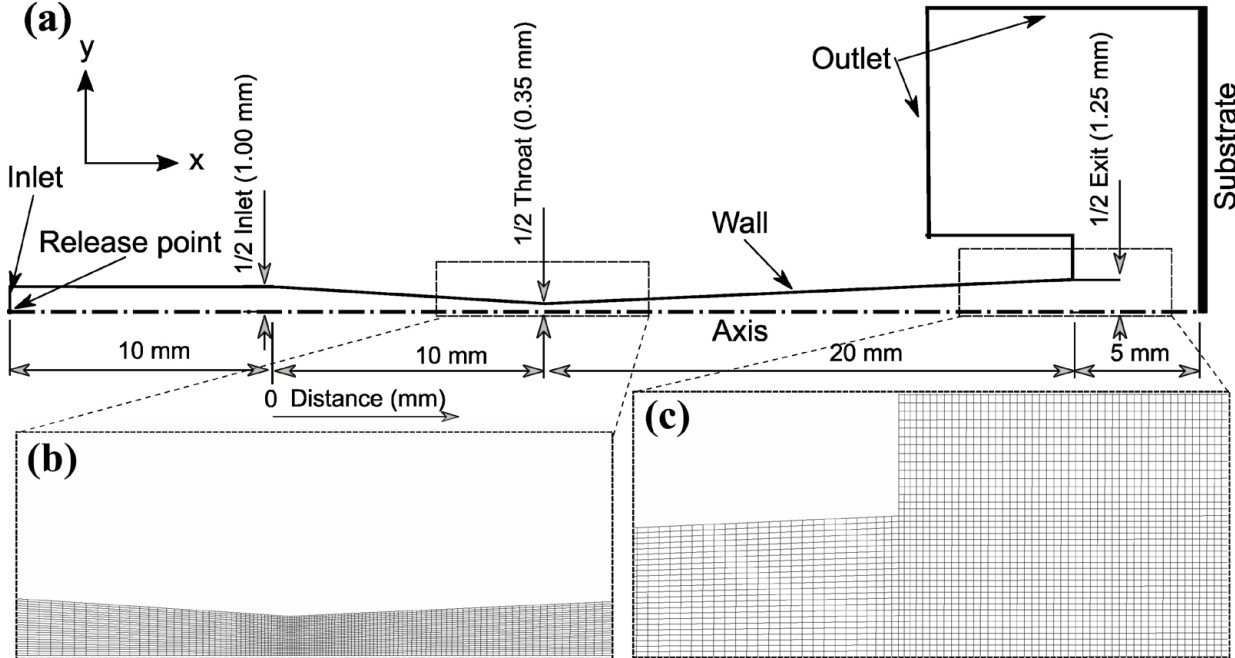

**Figure 2.** (**a**) Schematic diagram of the computational domain and the boundary conditions, with an enlarged view of mesh at (**b**) nozzle throat, and (**c**) nozzle exit.

The gas was taken as an ideal and compressible fluid, and a coupled implicit method based on the density was used to solve the flow field. For simulation accuracy, the shear-stress transport k-ω model was employed to simulate the turbulence flow [10,30]. The

discrete phase model available in Fluent with Lagrangian one-way coupling was used to compute the acceleration of copper particles.

Furthermore, the heat transfer between the gas flow and copper particles was taken into account. The spherical particles were released at a point in the VCS gas stream 10 mm upstream of the nozzle inlet with an initial temperature of 300 K (see Figure 2a). The Stochastic-Tracking type model available in Fluent was used to consider turbulence effects, in which the Discrete Random Walk model can be employed to predict the particle distribution and velocity.

## 3. Results and Discussion

### 3.1. Effect of Gas Preheating Temperature on Particle Acceleration

For an ideal gas, a one-dimensional isentropic model provides a straightforward and easily calculable means to estimate gas flow properties [31]. In this isentropic mode, the Mach number at the nozzle axis is solely dependent on nozzle size (expansion ratio) and gas species. When considering a specific nozzle and a defined working gas, elevating the inlet gas temperature emerges as the most effective method to significantly enhance the velocity characteristics of the flow field [32]. This approach has found widespread application in augmenting the quality of coatings in cold spraying, especially for materials with powders possessing a higher critical velocity [33,34]. Consequently, the inlet gas temperature is of considerable significance in the preparation of metal films via VCS.

Figure 3a–d illustrate the contour plots of gas flow velocity in four scenarios characterized by distinct inlet gas pressures (293 K, 423 K, 523 K, and 623 K). In addition, Figure 3e,f present plots depicting the gas flow velocity and temperature profiles along the nozzle's central axis as functions of the distance from the nozzle inlet. Remarkably, it becomes evident that, with increasing inlet gas temperature, the gas flow velocity distributions in these four cases share a similar pattern. As anticipated, the gas flow velocity is initially low at the inlet and increases towards the nozzle throat. The most significant acceleration in gas flow velocity occurs primarily at the nozzle throat, where the transition from subsonic to supersonic velocity takes place. In all four cases, the flow pressure at the nozzle exit remains higher than the chamber pressure (300 Pa) (refer to Table 2), ensuring the continued increase in supersonic flow velocity. As depicted in Figure 3f, the rise in gas flow velocity correlates with a proportional decrease in gas temperature. This phenomenon has also been observed in cold spraying applications [35–37]. Upon reaching the substrate, the supersonic flow experiences a sharp transformation, with the gas flow velocity plummeting from supersonic to zero.

Conversely, as the gas flow velocity decreases, the gas temperature increases rapidly in accordance with the declining flow velocity. In both cold spraying [38,39] and VCS [10,30], a bow shock with a low-velocity, high-density region commonly forms near the substrate's surface. As the inlet gas temperature escalates from 293 K to 623 K, the gas temperature at the bow shock increases from 309 K to 671 K. Concurrently, the thickness of the bow shock exhibits a slight increment, rising from 0.90 mm to 1.15 mm. These fluctuations in flow temperature and bow shock thickness significantly impact particle interactions, particularly for small particles, a subject we will delve into later.

The gas flow velocity and gas temperature demonstrate a conspicuous upward trend as the inlet gas temperature rises. This trend aligns with the outcomes predicted by isentropic theory (see Table 2). As the inlet gas temperature elevates from 293 K to 623 K, the simulated gas flow velocity at the nozzle exit's center surges from 1572 m/s to 2136 m/s. Due to the influence of viscous effects in the flow, the simulated gas flow velocities in all four cases are somewhat lower than those predicted by the isentropic theory. Consequently, as illustrated in Figure 4a, the Mach numbers along the nozzle's central axis in these four cases fall short of the Mach numbers calculated using the isentropic theory.

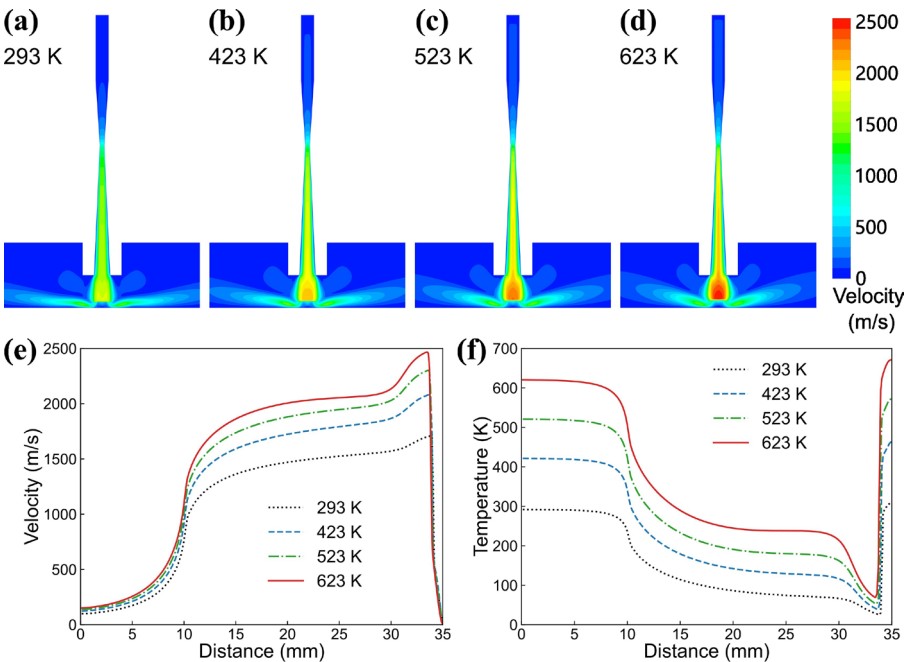

**Figure 3.** CFD gas flow velocity results with different inlet gas temperatures of (**a**) 293 K, (**b**) 423 K, (**c**) 523 K, and (**d**) 623 K. And changes of (**e**) velocity and (**f**) temperature of flow along the nozzle centerline.

**Table 2.** A comparison of CFD simulation results and isentropic nozzle theory for four inlet gas temperatures.

| Inlet Gas Temperature (K) | Velocity at Nozzle Exit Center (m/s) | | Maximum Flow Velocity (m/s) | Pressure at Nozzle Exit Center (Pa) | |
|---|---|---|---|---|---|
| | Isentropic | Simulation | | Isentropic | Simulation |
| 293 | 1668 | 1572 | 1709 | 412 | 875 |
| 423 | 2004 | 1868 | 2082 | 412 | 1082 |
| 523 | 2228 | 2031 | 2302 | 412 | 1208 |
| 623 | 2432 | 2136 | 2468 | 412 | 1295 |

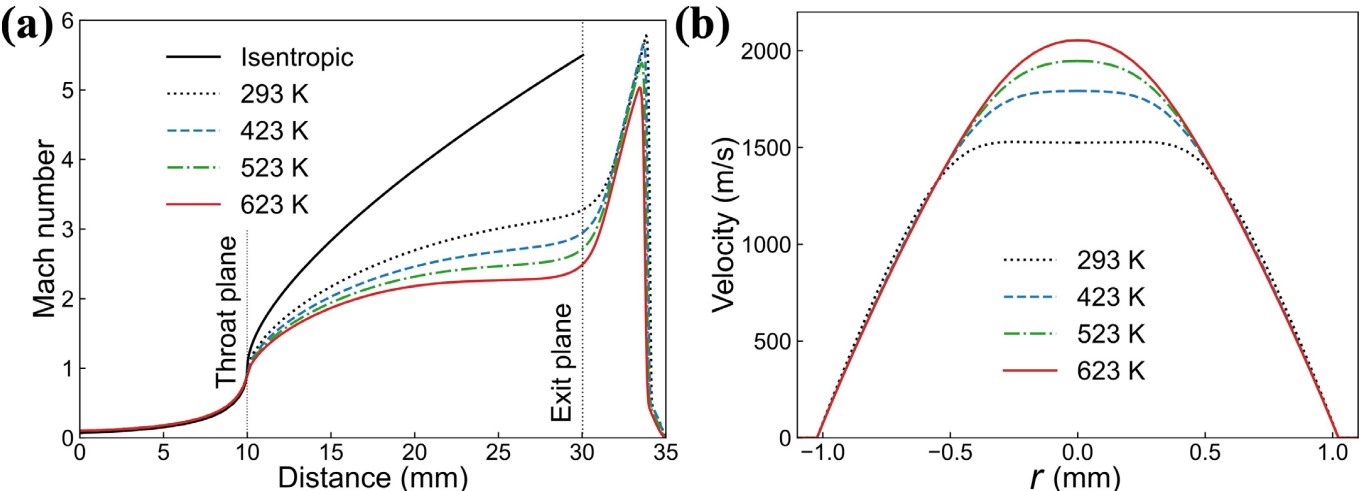

**Figure 4.** (**a**) Mach number along the nozzle center line and (**b**) gas flow velocity along the radial line 5 mm upstream from the nozzle exit plane.

Furthermore, as the inlet gas temperature increases, the Mach number inside the nozzle experiences a contrasting decrease. Concurrently, the boundary layer near the nozzle's inner wall thickens, as depicted in Figure 4b. Given the typical usage of slit nozzles or small-sized de Laval nozzles in VCS, the boundary layer is a critical factor that cannot be disregarded. Therefore, it is advisable to employ a near-wall model, as opposed to a wall function, to address the near-wall region in CFD simulations. Considering the chamber pressure (300 Pa) near the gas pressure at the nozzle exit center, as calculated using the isentropic theory (441 Pa), the maximum simulated gas velocity in all four cases closely aligns with that predicted by the isentropic theory.

Figure 5 illustrates the variation in the mean impact velocity and mean impact temperature of copper particles as a function of particle size. These particles experience two distinct stages: an acceleration stage inside the nozzle and a subsequent deceleration stage within the bow shock. Despite the rapid acceleration of small particles by the carrier gas, their velocity experiences a significant reduction upon passing through the bow shock due to their limited inertia. Consequently, there exists an optimal particle size that yields the highest particle impact velocity. This phenomenon is also observed in cold spraying simulations [40]. Notably, as the inlet gas temperature increases, the optimal particle size for achieving maximum particle velocity undergoes a slight increase. This can be attributed to the heightened gas stagnation pressure and gas density within the bow shock, leading to a more pronounced deceleration effect on smaller particles.

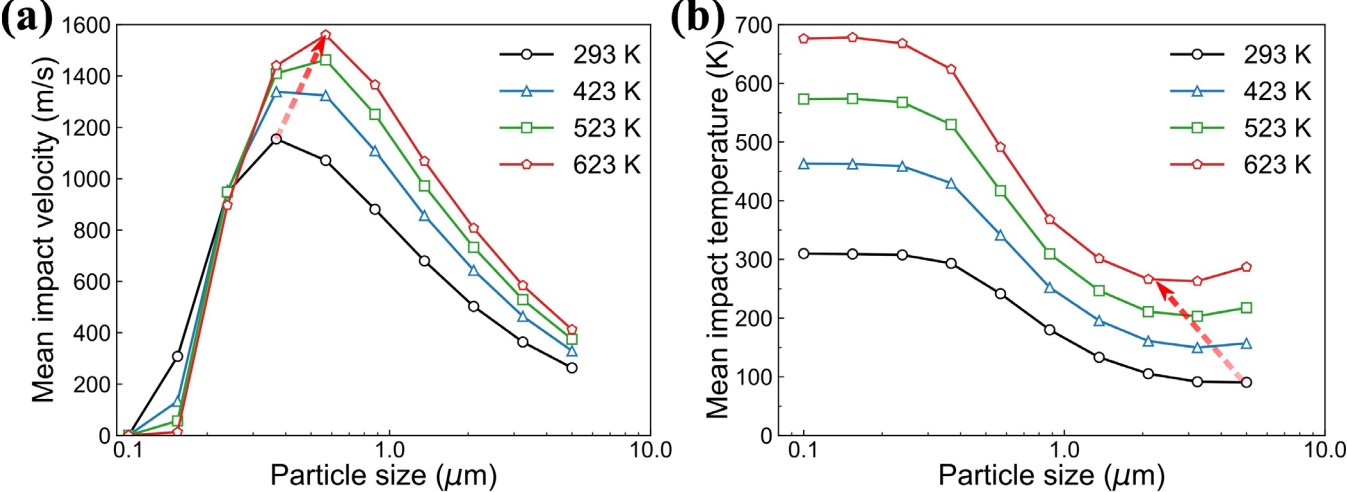

**Figure 5.** (**a**) Particle mean impact velocity and (**b**) mean impact temperature for different particle diameters at gas temperatures from 293 K to 623 K. (The arrows represent the variation in particle size corresponding to the lowest impact temperature.)

Furthermore, when the inlet gas temperature is raised, particles smaller than 0.2 μm exhibit lower particle impact velocities. This behavior is because smaller particles, possessing lower mass, can more readily respond to changes in gas temperature. Consequently, these small particles (<0.2 μm) attain an impact temperature very similar to the gas temperature of the bow shock (as depicted in Figure 5b). In contrast, the temperature of slightly larger particles (>2 μm) decreases with the gas temperature inside the nozzle. However, their brief time of residence within the bow shock is insufficient to acquire a significant amount of heat. Consequently, there exists a particle size corresponding to the lowest particle impact temperature. In summary, elevating the gas temperature before particle ejection can significantly enhance both the particle impact velocity and impact temperature, thereby promoting increased deformation during particle collisions.

## 3.2. Microstructure and Electrical Properties of Copper Films Deposited at Different Gas Temperatures

To investigate the influence of gas temperature on the preparation of VCS-deposited copper films, experiments were conducted at gas temperatures of 150 °C, 250 °C, and 350 °C with gas preheating, as well as at 20 °C without gas preheating. Figure 6 presents the surface and cross-sectional morphology of the copper films prepared on the alumina substrate at these four different temperatures. Remarkably, even without gas preheating, the copper film can be successfully deposited onto the alumina substrate. However, numerous spherical copper particles with insufficient deformation and noticeable gaps between them are evident on the film surface in Figure 6a,b. Without gas preheating, the copper particles lack the necessary impact velocity, resulting in insufficient deformation of the copper particles, making it difficult to achieve effective bonding. In the case of deposition at a gas preheating temperature of 150 °C, as depicted in Figure 6c,d, the number of undeformed particles on the film surface decreases, although gaps between the particles are still observable. With deposition at a gas preheating temperature of 250 °C (Figure 6e,f), the number of undeformed particles and gaps between particles on the film surface further diminishes, signifying a notable improvement in copper particle adhesion. When the gas preheating temperature reaches 350 °C (Figure 6g,h), the particles' outlines and the gaps between them are hardly discernible on the film surface. In other words, the copper particles undergo substantial flattening through plastic deformation and tightly adhere to form a dense copper film.

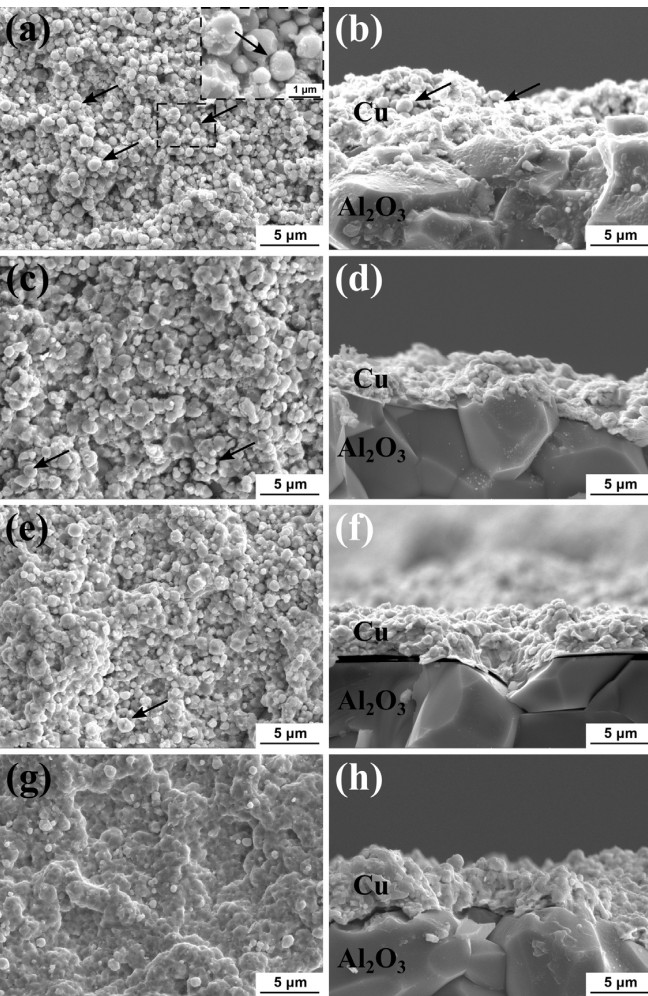

**Figure 6.** Microstructure morphologies of copper films deposited on Al$_2$O$_3$ substrates at gas preheating temperatures of (**a,b**) 20 °C, (**c,d**) 150 °C, (**e,f**) 250 °C, and (**g,h**) 350 °C. Black arrows mark undeformed copper particles.

From the simulation results in Section 3.1, the microstructural differences observed can be attributed to two key factors. Firstly, higher gas temperatures increase particle velocity during VCS, imparting greater kinetic energy to the particles and leading to more pronounced plastic deformation. This phenomenon parallels observations made during cold spray [20,21]. Secondly, elevated gas temperatures facilitate easier particle deformation [41,42]. In summary, increasing the gas temperature proves more conducive to the preparation of a dense VCS-deposited copper film.

Figure 7 presents the electrical resistivity of the copper films as a function of gas preheating temperature. The electrical resistivity decreases as the gas temperature increases. Specifically, at a gas preheating temperature of 20 °C, the resistivity of copper film exhibits its highest value, approximately $1.3 \times 10^{-6}$ Ω·m. This elevation in resistivity is attributed to the low particle binding and the presence of gaps between the particles. Notably, as the gas preheating temperature increases, the resistivity significantly decreases, eventually reaching approximately $4.4 \times 10^{-8}$ Ω·m at a gas preheating temperature of 350 °C. This value is only about 2.6 times that of bulk copper ($1.7 \times 10^{-8}$ Ω·m) [43], surpassing the previously reported AD copper film [17]. These results suggest that VCS-deposited copper films hold promise for future applications in ceramic surface metallization.

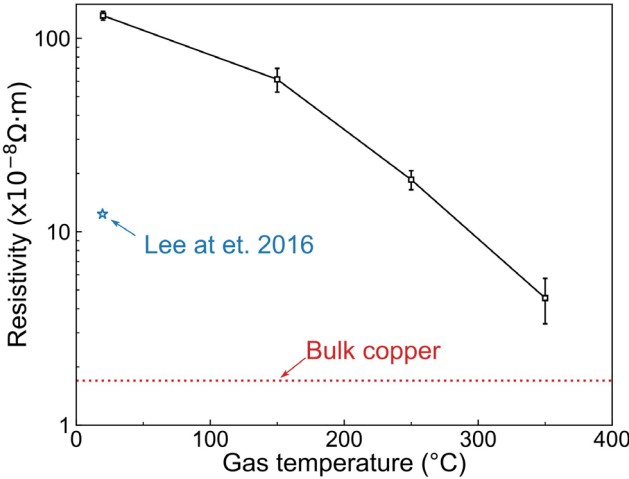

**Figure 7.** Electrical resistivity of copper films deposited at different gas temperatures. (The dotted line is the resistivity of bulk copper, and asterisk data comes from the literature [17]).

The electrical resistivity of VCS-deposited copper films is closely linked to both particle bonding and film compactness. Higher gas temperatures (higher than 250 °C) result in increased particle impact velocity and temperature, which, in turn, enhance the microcontacts among particles, ultimately leading to the formation of a denser copper film. Additionally, the presence of $Cu_2O$ and $CuO$ oxides in copper films plays a crucial role in elevating the electrical resistivity of the copper film above that of bulk copper [17]. Consequently, future research should focus on controlling the oxygen content of sub-micron copper powder and preventing oxidation during particle deposition to further optimize the electrical properties of these films.

### 3.3. Copper Films Grown on Alumina Substrates

To investigate the growth of the copper films during the VCS process, the film thickness was measured after each scan. Figure 8 displays the film thickness as a function of the number of scans. When performing a single scan, as shown in Figure 9a, the copper particles completely cover the surface of the alumina substrate, forming a continuous copper thin film with an initial thickness of approximately 4.7 μm. Subsequently, the film thickness exhibits a linear increase as the number of scans increases. The rate of thickness growth reaches approximately 4.3 μm per scan, resulting in a high deposition rate for a $10 \times 10$ mm$^2$ area, equivalent to 1.03 μm per minute. By the time 15 scans are completed,

the thickness of the copper film reaches approximately 63 µm. In essence, vacuum cold spraying can be employed to produce copper thin films in the range of several microns in thickness, and thicker copper films to the order of tens of microns or even more, by increasing the number of scans.

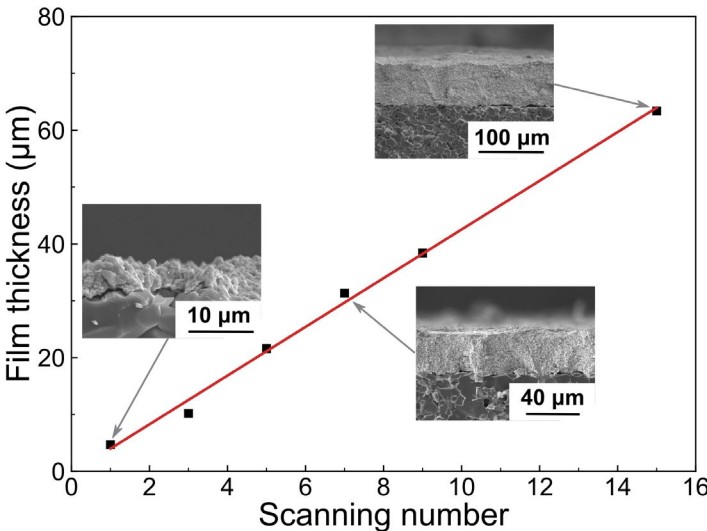

**Figure 8.** Thickness of copper films deposited at different scanning numbers.

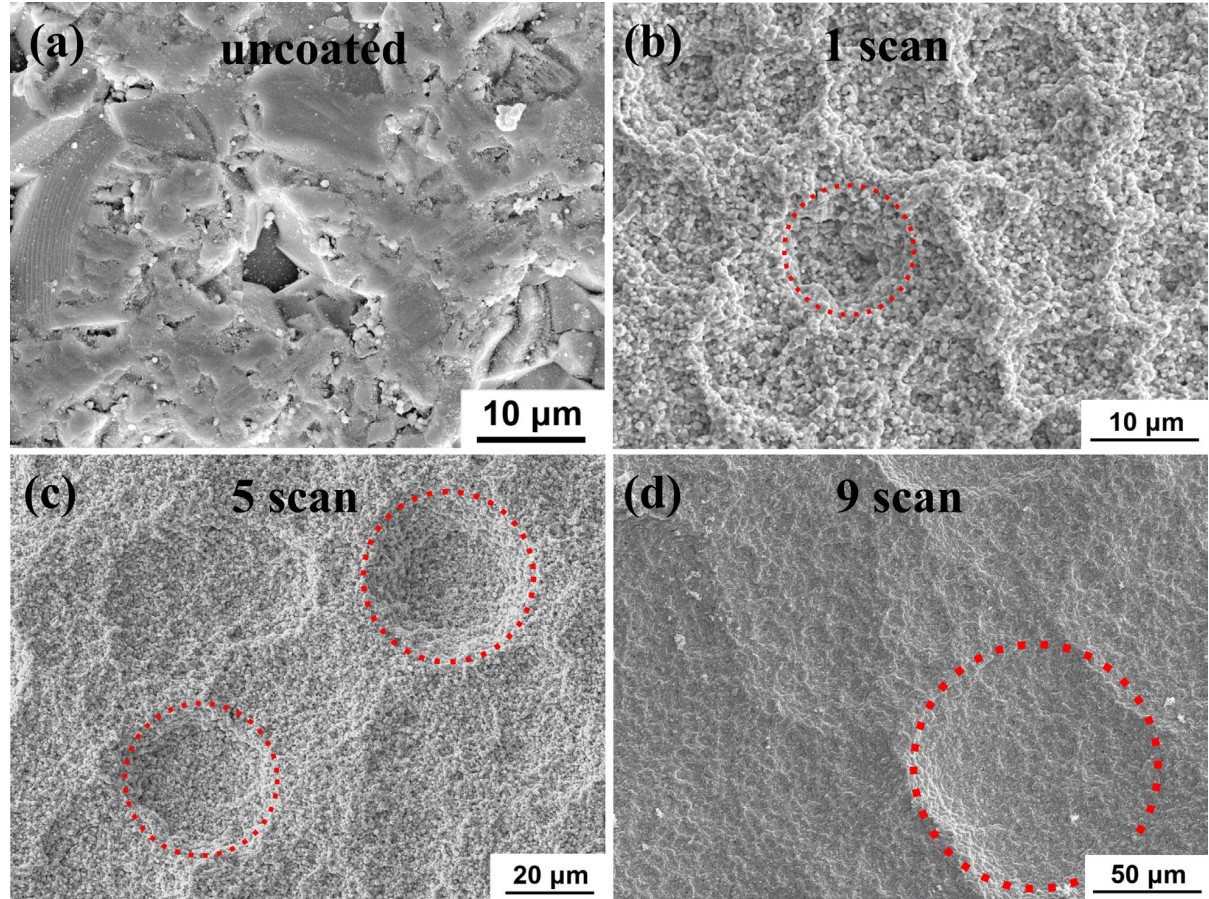

**Figure 9.** Surface topography of copper films on an alumina substrate with different scanning numbers: (**a**) uncoated, (**b**) 1 scan, (**c**) 5 scans and (**d**) 9 scans. The red dot circle indicates one-shaped pits.

Figure 9 presents surface microstructure images of the copper films on the alumina substrate as the number of scans increases. While a single scan results in complete coverage of the alumina surface by copper particles, the surface of the copper films exhibits significant roughness, accompanied by the presence of cone-shaped pits (highlighted by red dotted circles in Figure 9a). This phenomenon arises due to the thin nature of the coating, which inherits the irregularities of the substrate surface. As the number of scans increases, the thickness of the copper film steadily increases, and the edges of the cone-shaped pits expand outward, leading to an increase in their diameter. As the film thickness increases from 5 μm to 38 μm, the diameter of the cone-shaped pits expands from approximately 11 μm to around 91 μm, as shown in Figure 10. With further increases in film thickness, the cone-shaped pits gradually vanish, and their edges become increasingly difficult to discern.

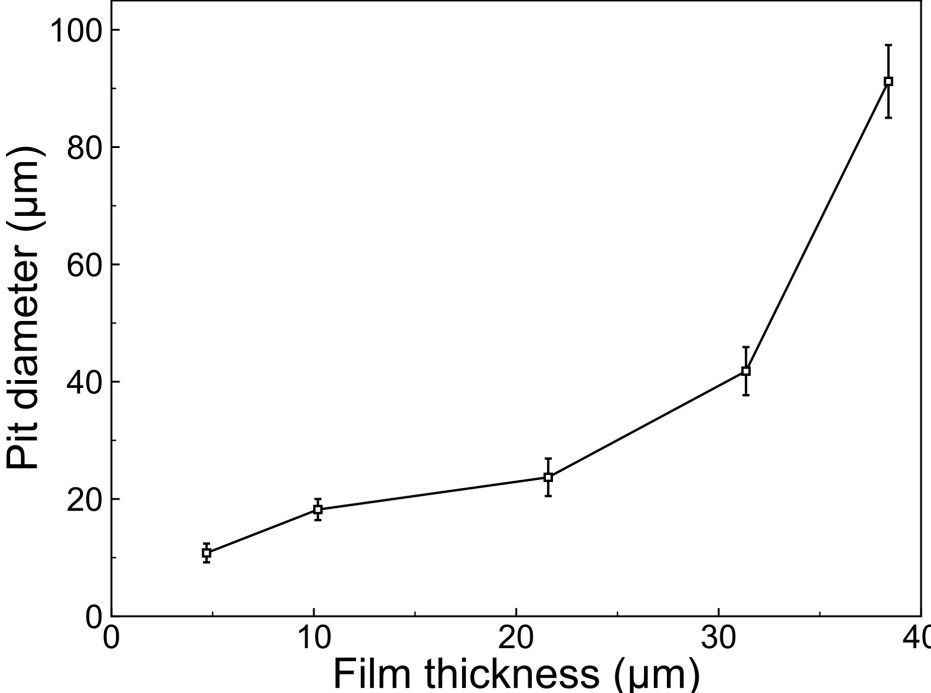

**Figure 10.** Diameter of the cone-shaped pits on copper films with different film thicknesses.

The cross-sectional view of one of these cone-shaped pits, as shown in Figure 11a, confirms that it originates from a pit on the substrate's surface. Furthermore, the pit diameter undergoes more pronounced changes with the thickness of the copper films, as depicted in Figure 10. As the pit diameter increases, the pit walls become flatter, ultimately resulting in the gradual disappearance of these larger pits. Consequently, observing large pits on the surface of copper films with a thickness exceeding 40 μm becomes challenging. Figure 11b displays the SEM image of the peeled surface of the copper film from the alumina substrate. This image reveals that small pits on the substrate surface can be adequately filled by an individual or a limited number of copper particles (highlighted in blue). However, for slightly larger pits, multiple particles are required for filling (highlighted in red). Furthermore, larger pits, such as those within the red circle in Figure 9a and the pits in Figure 11a, result in the persistence and evolution of cone-shape pits. Mechanical interlocking is the primary factor contributing to the adhesion of metal coatings to ceramic substrates. The presence of certain irregularities on the substrate surface further enhances the substrate's ability to achieve a robust bond with the copper film [17,44,45].

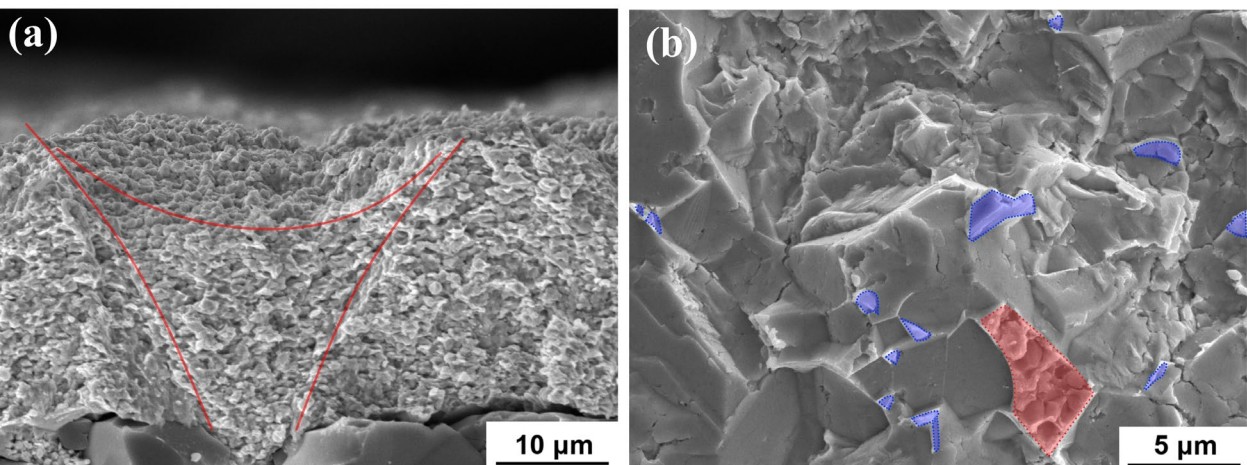

**Figure 11.** (**a**) Cross-sectional SEM image of the cone-shape pit on the copper film, and (**b**) SEM image of the peeled surface of the copper film from the substrate. The blue highlighting represents small pits filled with a single or a limited number of copper particles, while the red highlighting indicates large pits filled with multiple copper particles.

Figure 12 provides a schematic representation of copper film growth on a rough alumina substrate. The substrate exhibits numerous small pits measuring several hundred nanometers in size, along with larger pits of a few microns, as depicted in Figure 12a. Unlike the deposition mechanism observed in brittle particles such as ceramics, which involves the reduction of crystallite size during the VCS process [7,46], ductile metals primarily undergo plastic deformation during deposition [17,19,22]. When copper particles collide with the high-hardness alumina substrate, they experience high stress levels, resulting in substantial plastic deformation. Consequently, the valleys in the rough alumina substrate become covered with deformed copper particles, forming a mechanical interlock between the copper films and the alumina substrate (Figure 12b).

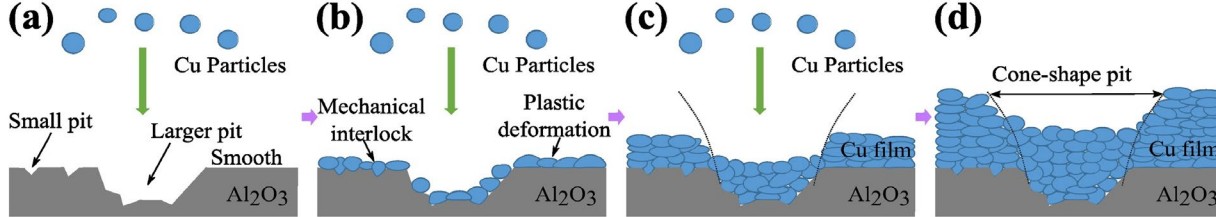

**Figure 12.** The schematic diagram of the copper film grown on a rough alumina substrate. (**a**) Uncoated $Al_2O_3$ substrate, (**b**) initial deposition process, (**c**) subsequent deposition process and (**d**) deposition completion resulting in the formation of a copper film.

However, there are some larger pits whose sizes exceed that of individual copper particles. While copper particles are also deposited inside these pits, they tend to remain within the pits and become incorporated into the film (Figure 12c). It has been reported that the impact angle plays a significant role in the deposition efficiency of copper in the cold spray process, with lower deposition efficiency observed at smaller impact angles [47]. As the thickness of the copper films increases, the cone-shaped pits expand outward due to the lower deposition efficiency of copper particles on the sidewalls of these pits (Figure 12d). Once the thickness of the copper films reaches a certain threshold, the side walls of the cone-shaped pits tend to become flatter, eventually leading to the gradual disappearance of these pits. Consequently, there exists a self-adaptive repair mechanism when depositing metal films on a rough substrate using VCS.

## 4. Conclusions

In this study, the results from CFD simulations reveal that even in a chamber with pressure below 300 Pa, the bow shock near the substrate still exerts an influence on the impact velocity of ultrafine particles. Moreover, there exists an optimal particle size that yields the highest particle impact velocity. Concurrently, the significant impact of gas temperature on particle impact temperature and impact velocity is confirmed. Compared to room temperature conditions, appropriately increasing the gas temperature enhances particle velocity and softens the particles, leading to substantial particle deformation and the formation of a dense copper film. Additionally, copper films were successfully fabricated on alumina substrates using the vacuum cold spray (VCS) process. The electrical resistivity of the copper films decreases as the gas preheating temperature increases. Specifically, the electrical resistivity decreases from $1.3 \times 10^{-6}$ $\Omega\cdot$m to $4.4 \times 10^{-8}$ $\Omega\cdot$m as the gas temperature increases from 20 °C to 350 °C. In summary, vacuum cold spraying demonstrates potential for applications in ceramic surface metallization.

Furthermore, the thickness of the copper films exhibits a linear increase with an increase in the number of scans. This implies that VCS can be employed to produce thin metal films in the range of several microns and thicker metal films ranging from tens of microns to even thicker layers. When copper films are prepared on rough alumina substrates, cone-shaped pits are observed on the film surface. As the copper films grow, the pit diameter increases and eventually disappears. It has been confirmed that the evolution of these pits is primarily influenced by the impact angle. On the high-hardness, rough alumina substrate, copper particles undergo plastic deformation and fill the valleys on the rough surface, creating a mechanical interlocking effect. For larger pits whose sizes exceed the particle size, their profiles are transferred to the film surface and tend to flatten as the film thickness increases. Consequently, there exists a self-adaptive repair mechanism when depositing metal films on rough ceramic substrates using the VCS process.

**Author Contributions:** Conceptualization, K.M. and C.-X.L.; Methodology, K.M., Q.-F.Z., H.-Y.Z. and C.-X.L.; Validation, K.M. and Q.-F.Z.; Formal analysis, K.M.; Investigation, K.M. and Q.-F.Z.; Resources, C.-J.L.; Data curation, Q.-F.Z.; Writing—original draft, K.M.; Writing—review & editing, H.-Y.Z. and C.-J.L.; Visualization, K.M.; Supervision, C.-X.L.; Funding acquisition, C.-X.L. All authors have read and agreed to the published version of the manuscript.

**Funding:** This research was funded by the State Key Laboratory for the Mechanical Behavior of Materials, Xi'an Jiaotong University. All the samples were fabricated and characterized in the thermal spray laboratory (TSL-XJTU).

**Institutional Review Board Statement:** Not applicable.

**Informed Consent Statement:** Not applicable.

**Data Availability Statement:** Not applicable.

**Conflicts of Interest:** The authors declare no conflict of interest.

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
