# Peer review of "Influence of the Gas Preheating Temperature on the Microstructure and Electrical Resistivity of Copper Thin Films Prepared via Vacuum Cold Spraying"

_coatings, doi:10.3390/coatings13111870_

Round 1
Reviewer 1 Report
Comments and Suggestions for Authors
An interesting and well-written article, but some minor issues can be found:
1. Line 153 - seems that some text is missed.
2. It would be good to add a SEM image of an uncoated sample surface.
3. More detailed description of samples could be added (manufacturer, methods of surface preparation for experiments - used as is, cleaned, baked)
Author Response
We sincerely thank you for your positive comments and kind suggestions. In the following, you can find our replies. All corrections are highlighted with red color which you can find easily in the revised text.
- Line 153 - seems that some text is missed.
Author's response: We are sorry for this error, and it has been modified.
- It would be good to add a SEM image of an uncoated sample surface.
Author's response: Thanks for your suggestion. The SEM image of uncoated sample surface has been added as Figure 9a in new manuscript
- More detailed description of samples could be added (manufacturer, methods of surface preparation for experiments - used as is, cleaned, baked)
Author's response: Thanks for your suggestion, and we have modified this part in the new manuscript.
Reviewer 2 Report
Comments and Suggestions for Authors
Dear Authors! I reviewed your submission "Influence of Gas Preheating Temperature on Microstructure and Electrical Resistivity of Copper Thin Films Prepared by Vacuum Cold Spraying". It is generally worth and suitable for publication, but after minor revision.
Line 45. "notable absence of comprehensive research focused on the 45 preparation of metal films using VCS" - not fully true - one can find thousands of publications and some of them do cover the topics. Actually, the Introduction is the weakest part of your paper. Why you mentioned MEMS and microelectronics - many microns (up to 40 in your case) is certainly other application. Please, improve.
Line 82 "is provided in " Table 1, I guess
Lines 152-153 - what is this? A fault I guess. Please, fix.
Line 214 "resulting in a low binding ratio among copper particles" - In fact, this argument is speculative. You did not study this aspect. Please, re-phrase.
Author Response
We sincerely thank you for your positive comments and kind suggestions. In the following, you can find our replies. All corrections are highlighted with red color which you can find easily in the revised text.
- Line 45. "notable absence of comprehensive research focused on the 45 preparation of metal films using VCS" - not fully true - one can find thousands of publications and some of them do cover the topics. Actually, the Introduction is the weakest part of your paper. Why you mentioned MEMS and microelectronics - many microns (up to 40 in your case) is certainly other application. Please, improve.
Author's response: Thanks for your valuable and kind suggestion. We have modified this part in the new manuscript. One of the primary advantages of vacuum cold spraying lies in its capability to produce thin films (<20 μm), which holds potential applications in the conductivity or heat conduction components of MEMS or microelectronics. This article addresses thicknesses of tens of micrometers only to illustrate that this method is also capable of producing thicker films. However, when it comes to preparing coatings of tens of micrometers, compared with the more efficient atmospheric cold spraying, VCS has no advantage.
- Line 82 "is provided in Table 1, I guess
Author's response: You are right, and it has been modified.
- Lines 152-153 - what is this? A fault, I guess. Please, fix.
Author's response: You are right, and it has been modified.
- Line 214 "resulting in a low binding ratio among copper particles" - In fact, this argument is speculative. You did not study this aspect. Please, re-phrase.
Author's response: Thanks for your valuable and kind suggestion. We have modified this part in the new manuscript.
“Without gas preheating, the copper particles lack the necessary impact velocity, resulting in insufficient deformation of the copper particles, making it difficult to achieve effective bonding.”
Reviewer 3 Report
Comments and Suggestions for Authors
This paper describes the influence of gas preheating temperature for copper thin films by vacuum cold spraying and presenting a film formation process model. While the subject matter of this paper is interesting and important to the readers on ceramic surface metallization, some conditions for both simulation and experiment are insufficient and unclear.
For example, the dimension (length) of the nozzle is incompatible between in Figure 2 and 3, 4, 35 mm in the former case and 30 mm in latter cases.
We recommend that the previous points be clarified and revised prior to publication in Coatings.
In addition, please consider the following points.
(1) Line 18, Line 341: The nth power should be superscript.
(2) Line 82: The sentence ends in the middle and does not make sense.
(3) Line 114: In Fig. 2 b and c, please indicate the figure so that the reader can see the correspondence with the shape of a.
(4) Line 177: Make sure that the reader can see the correspondence between the X-axis origin in Figure 4 and the nozzle start point in Figure 2a, or where the origin corresponds to it. Also, check again that the outlet is 30 mm from the nozzle inlet.
(5) Line 231: In figure 6 a, it is hard to recognize the spherical copper particle with insufficient deformation. It is better to add the enlarged one.
(6) Line 258: Specify the film deposition conditions (gas temperature) of the VCS process.
Author Response
We sincerely thank you for your positive comments and kind suggestions. In the following you can find our replies. All corrections are highlighted with red color which you can find easily in the revised text.
- For example, the dimension (length) of the nozzle is incompatible between in Figure 2 and 3, 4, 35 mm in the former case and 30 mm in latter cases.
Author's response: We are sorry for this error, and it has been modified in Figure 2.
- In addition, please consider the following points.
(1) Line 18, Line 341: The nth power should be superscript.
(2) Line 82: The sentence ends in the middle and does not make sense.
(3) Line 114: In Fig. 2 b and c, please indicate the figure so that the reader can see the correspondence with the shape of a.
(4) Line 177: Make sure that the reader can see the correspondence between the X-axis origin in Figure 4 and the nozzle start point in Figure 2a, or where the origin corresponds to it. Also, check again that the outlet is 30 mm from the nozzle inlet.
(5) Line 231: In figure 6 a, it is hard to recognize the spherical copper particle with insufficient deformation. It is better to add the enlarged one.
(6) Line 258: Specify the film deposition conditions (gas temperature) of the VCS process.
Author's response: Thanks for your valuable and kind suggestion. We have modified them in the new manuscript.
(1) They has been modified.
(2) It has been modified.
(3-4) Figure2 has been redrawn.
(5) Figure6a has been modified.
(6) It has been modified.
Reviewer 4 Report
Comments and Suggestions for Authors
Re.: Coatings, manuscript coatings-2661100-peer-review-v1.pdf
Title: Influence of Gas Preheating Temperature on Microstructure and Electrical Resistivity of Copper Thin Films Prepared by Vacuum Cold Spraying
Authors: Kai Ma, Qing-Feng Zhang, Hui-Yu Zhang, Chang-Jiu Li, Cheng-Xin Li
General Statement
In the paper results of complex: theoretical numerical and experimental investigations of the Vacuum Cold Spray deposition of Cu layer onto alumina ceramic structure are presented and discussed. While the numerical analysis is focused on the phenomena of transient ballistics - heat and mass exchange in the gas stream between copper powder particles, the experimental part concerns the study of the structure of the produced layers. Both part complement each other. I have no reservations about the compatibility of this topic with the journal's profile.
Analyzing the content of the work, I notice that the methods used are known. However, the work is original and meets the requirement of novelty in the scope of the specific research case analyzed. I highly appreciate the substantive level of the work. In general the paper is also well organized and in terms of scientific description sounds good. A few minor editing errors and some inaccuracies would not prevent the work from being directly recommended for publication if it were not for one issue requiring clarification or discussion. It concerns a small but important substantive detail. Any explanation does not affect the presented results and content, but may clarify the perception of the work. Details are presented below. Because of that I recommend the work for publication after discussion, i.e. after major revision.
Major Comment
Re.: lines 105-106
The above indicated problem concerns one way coupling assumption applied in CFD modelling. One-way coupling is a reasonable assumption if a powder mass flow is about one range lower than the carrier gas mass flow. I am afraid that when using a light gas such as helium for the actual process, maintaining the proportions would significantly extend the time of the actual process. Could the authors provide data on powder feeding rate? This information will allow you to keep some distance when assessing the results obtained.
Minor Suggestions
Re.: lines 18, 32, 61-64, 82, 153-154; five coments in total except the one mentioned above
While reading the work, I marked the fragments of the text to which the comment concerned. The manuscript with markings is attached.

I have no additional comments.
Author Response
We sincerely thank you for your positive comments and kind suggestions. In the following you can find our replies. All corrections are highlighted with red color which you can find easily in the revised text.
1.Re.: lines 105-106
The above indicated problem concerns one way coupling assumption applied in CFD modelling. One-way coupling is a reasonable assumption if a powder mass flow is about one range lower than the carrier gas mass flow. I am afraid that when using a light gas such as helium for the actual process, maintaining the proportions would significantly extend the time of the actual process. Could the authors provide data on powder feeding rate? This information will allow you to keep some distance when assessing the results obtained.
Author's response: Thanks for your valuable and kind suggestion. In our experimental setup, the powder feed rate is approximately 0.15g/min, with a powder mass flow rate less than 1/10 of the gas mass flow rate. Given the complexity of bidirectional coupling, we are currently focusing solely on one-way coupling. We agree with your viewpoint that an excessively high powder feed rate can affect the gas flow field. In our future work, we will investigate bidirectional coupling and explore the impact of powder mass flow rate on gas flow distribution.
- Re.: lines 18, 32, 61-64, 82, 153-154; five coments in total except the one mentioned above.
Author's response: Thanks for your valuable and kind suggestion. We have modified them in the new manuscript.
(1) line 18: It has been modified.
(2) line 32: Refs have been updated.
(3) line 61-64: First, CFD modeling was used to obtain the influence of gas preheating on the gas flow field and particle impact velocity. Provide theoretical support and parameter suggestions for subsequent experiments.
(4) line 82: It has been modified.
(5) line 153-154: It has been modified.
Reviewer 5 Report
Comments and Suggestions for Authors
The manuscript “ Influence of Gas Preheating Temperature on Microstructure and Electrical Resistivity of Copper Thin Films Prepared by Vacuum Cold Spraying ”, by Kai Ma, Qing-Feng Zhang, Hui-Yu Zhang, Chang-Jiu Li, Cheng-Xin Li, highlights an interesting design approach, by using both, experimental research, combined with computer science tool. In the manuscript, the authors examine the evolution of cone-shaped pits on the surface of copper films prepared on rough substrates, along with CFD simulation in order to investigate the impact of gas preheating temperature on particle impact velocity and temperature. The paper can be published after minor changes. My comments are below.
1. Line 77: Please insert the purity of the used He gas.
2. Line 78: Please insert the specification of the used heating device (company name, technical parameters used in the measurements, etc.). If it’s an oven which heats in steps, etc., please specify it.
3. Line 80: What type of substrate was used to deposit the copper films? Please specify it inside the manuscript.
4. Line 82: Please finish the sentence: “… VCS process is provided in…”
5. Line 96: What mesh type have you used? The triangle one? How did you increment the mesh at the boundaries?
6. Line 108: Did you have constraints related to the size of the used Cu particles considered in your simulations?
7. Lines 153-154: You have some unfinished sentences between branches.
8. Lines 275-276: Please close the branches.
Author Response
We sincerely thank you for your positive comments and kind suggestions. In the following, you can find our replies. All corrections are highlighted with red color which you can find easily in the revised text. Detailed replies are as follows:
- Line 77: The purity of He gas has been added.
- Line 78: The heating device was home-made in our laboratory.
- Line 80: The substrate was alumina slides (12 mm × 12 mm × 2 mm) with a surface roughness of Ra 1.0 μm.
- Line 82: It has been modified.
- Line 96: We used structured mesh, and our meshing is very fine and verified for mesh independence.
- Line 108: Yes, when adding copper particles using the discrete term model, copper particles with different diameters were set.
- Lines 153-154: It has been modified.
- Lines 275-276: It has been modified.
Round 2
Reviewer 3 Report
Comments and Suggestions for Authors
This paper reports the influence of gas preheating temperature for copper thin films by vacuum cold spraying and presenting a film formation process model. The revised manuscript is properly revised to be suitable for publication in its current form.